# Identification of Conservation Priority Areas and a Protection Network for the Siberian Musk Deer (*Moschus moschiferus* L.) in Northeast China

**DOI:** 10.3390/ani12030260

**Published:** 2022-01-21

**Authors:** Chao Zhang, Yuwei Fan, Minhao Chen, Wancai Xia, Jiadong Wang, Zhenjie Zhan, Wenlong Wang, Tauheed Ullah Khan, Shuhong Wu, Xiaofeng Luan

**Affiliations:** 1School of Ecology and Nature Conservation, Beijing Forestry University, No. 35 Tsinghua East Road, Haidian District, Beijing 100083, China; zc21192@163.com (C.Z.); fanyuwei2021@163.com (Y.F.); Demominhao@gmail.com (M.C.); wangjiadong_2019@163.com (J.W.); a472705335@163.com (Z.Z.); wangwenlong1016@163.com (W.W.); eco.tauheed@hotmail.com (T.U.K.); 2Key Laboratory of Southwest China Wildlife Resources Conservation (Ministry of Education), China West Normal University, Nanchong 637009, China; xiawancai2013@163.com

**Keywords:** conservation priority areas, *Moschus moschiferus*, potential connectivity corridors, protection network

## Abstract

**Simple Summary:**

Siberian musk deer (*Moschus moschiferus* L.) populations and their habitat decreased significantly over the past decades due to unsustainable, long-term hunting and deforestation. To support the species’ conservation, we assessed its potential distribution, conservation priority areas, core patch fragmentation, and potential connectivity corridors in Northeast China. We concluded that large areas of a high-quality *M. moschiferus* habitat with low fragmentation and low human influence remain in the northern Greater Khingan Mountains. In contrast, a habitat in the Lesser Khingan Mountains and the Changbai Mountains was highly fragmented and highly influenced by human activity. The Greater Khingan Mountains offer a suitable habitat for the recovery of *M. moschiferus* populations in Northeast China. These findings offer recommendations to support the Chinese government’s goal of establishing protected area systems with national parks in Northeast China.

**Abstract:**

Species conservation actions are guided by available information on the biogeography of the protected species. In this study, we integrated the occurrence data of Siberian musk deer (*Moschus*
*moschiferus* L.) collected from 2019 to 2021 with species distribution models to estimate the species’ potential distribution in Northeast China. We then identified conservation priority areas using a core-area zonation algorithm. In addition, we analyzed core patch fragmentation using FRAGSTATS. Lastly, we identified potential connectivity corridors and constructed a potential protection network based on the least-cost path and the circuit theory. The results showed concentrations of *M. moschiferus* in the northern Greater Khingan Mountains, the southeastern Lesser Khingan Mountains, and the eastern Changbai Mountains, with a potential distribution area of 127,442.14 km^2^. Conservation priority areas included 41 core patches with an area of 106,306.43 km^2^. Patch fragmentation mainly occurred in the Changbai Mountains and the Lesser Khingan Mountains. We constructed an ecological network composed of 41 core patches and 69 linkages for *M. moschiferus* in Northeast China. The results suggest that the Greater Khingan Mountains represent the most suitable area to maintain the stability of *M. moschiferus* populations in Northeast China. Considering the high habitat quality requirements of *M. moschiferus* and its endangered status, we propose that the Chinese government accelerates the construction of the Greater Khingan Mountains National Park and the Lesser Khingan Mountains National Park and enlarges the Northeast China Tiger and Leopard National Park to address the fragmentation of protected areas and the habitat of *M. moschiferus*.

## 1. Introduction

Musk deer (*Moschus* spp.) are important economic species, as the musk secreted from male preputial glands is a valuable ingredient in traditional eastern medicine and perfumes [1,2]. The high demand for musk in the pharmaceutical and perfume industries drove unsustainable and illegal long-term hunting for musk deer in Asia [3,4]. China used to have the largest musk deer populations worldwide, and the use of musk can be traced back to the Han Dynasty (202 BC to 220 AD) [5,6]. Musk deer hunting reached its peak in China in the 1970s when the international market value of musk reached USD 45,000 per kilogram [4,5]. It has been estimated that the wild musk deer population in China decreased by about 96% from the 1960s to the 1980s, and the populations in the Hebei and Guangdong Provinces have been extirpated [6,7,8]. As a reaction, the Chinese government listed musk deer as national first-class-protected animals in 2003. Although strict hunting bans have been implemented by the government to protect musk deer, their recovery is slow due to human-driven habitat degradation and fragmentation [4,5,7]. Habitat fragmentation has weakened the viability of isolated musk deer populations and has become a serious threat to the survival of musk deer [2,9].

The Siberian musk deer (*Moschus moschiferus* L.) is the most widely distributed species of the genus *Moschus*. It inhabits coniferous and broad-leaved mixed forests in Russia, Northeast China, Mongolia, Korea, and eastern Kazakhstan [4,5,10]. Two subspecies are found in China: *M. m. moschiferus* in the Greater Khingan Mountains, and *M. m. parvipes* in the Lesser Khingan Mountains and the Changbai Mountains. The International Union for Conservation of Nature (IUCN) classifies *M. moschiferus* as vulnerable, but notes that the species is declining more seriously than previously estimated and might qualify as endangered [4]. Understanding the geographic distribution of the species is pivotal in its conservation [11,12]. Within the last decades, many studies on the biogeography of *M. moschiferus* concentrated almost exclusively on Mongolia, Russia, and South Korea [3,13,14,15,16]. The distribution of *M. moschiferus* in Northeast China is a result of multiple forms of anthropogenic disturbance, but only a few studies in China have provided accurate biogeographical information on the species to support stakeholder decisions on its conservation [2,6,11]. An assessment report of the IUCN also noted that fundamental data on *M. moschiferus* in China are unavailable [4]. Therefore, there is an increasing and unmet need to evaluate the core distribution range of the species to support conservation efforts in China.

Constructing protected areas is the global key strategy for in-situ biodiversity conservation [17,18,19]. A total of 118,000 protected areas have been established in China as of 2019, which cover 18% of its land area and 4.6% of its sea area [20,21]. Furthermore, China launched one of the largest natural forest conservation programs globally, including a comprehensive ban on logging in natural forests, to protect biodiversity [14,22,23]. These efforts have resulted in a significant improvement of the natural environment and the populations of some endangered species, such as the giant panda (*Ailuropoda melanoleuca*) and Amur tiger (*Panthera tigris altaica*) [24,25]. However, the construction of protected area networks is flawed and lacks unified planning and coordination in China, which limits their effectiveness [26,27]. To overcome these limitations, the Chinese government is building a new national park-based system of protected areas for biodiversity conservation [26]. *M. moschiferus* inhabiting Northeast China is listed as a flagship species for the establishment of national parks under the China National Standard (ID: GB/T 39737-2020). Therefore, there is an urgent need to identify the potential connectivity corridors of this flagship species and to construct a protection network to establish protected area systems with national parks in Northeast China.

In this study, we integrated the occurrence data of *M. moschiferus* collected from 2019 to 2021 with an ensemble forecasting approach to assess the potential distribution of the species. We then identified conservation priority areas using a systematic conservation planning algorithm. In addition, we analyzed patch fragmentation of high-quality habitats in different biogeographic regions. Finally, we identified potential connectivity corridors and constructed a potential protection network based on the least-cost path and the circuit theory.

## 2. Materials and Methods

### 2.1. Study Area

The study area was located in Northeast China, including the Inner Mongolia Autonomous Region and Heilongjiang, Jilin, and Liaoning Provinces (38°42′–53°17′ N, 115°30′–135°06′ E). The region spans the warm-temperate, middle-temperate, and cold-temperate zones and covers the largest natural forest area in China [28,29]. The main mountains ranges include the Greater Khingan Mountains, Lesser Khingan Mountains, Zhangguangcai Mountains, Wanda Mountains, and Changbai Mountains, and the altitude ranges from 0 to 2691 m [30]. The diverse climate and unique geographical conditions form a complex ecosystem supporting many wildlife species, including the black-billed capercaillie (*Tetrao urogalloides*), lynx (*Lynx lynx*), and Amur tiger [30]. The map of Northeast China was obtained from the National Catalogue Service for Geographic Information (https://www.webmap.cn/, accessed on 7 September 2021) (Figure 1).

### 2.2. Data Collection

A total of 192 effective occurrence points of *M. moschiferus* were collected from 2019 to 2021 via interviews (*n* = 79), questionnaires (*n* = 28), field surveys (*n* = 28), and literature records (*n* = 57) (for records from literatures, see Appendix A) (Figure 1). To minimize errors, we cross-checked all occurrence records by comparison with Google Earth imagery to confirm the presence of a suitable habitat. Then, we removed duplicate records according to the movement range of *M. moschiferus* (6 km) to eliminate bias caused by clustered occurrences [4]. Ultimately, we obtained 176 occurrence points for *M. moschiferus* distribution modeling.

The selection of environmental variables is critical when using species distribution models (SDMs), and is ideally based on well-documented physiological and ecological variables [31,32]. We referred to previous research on *M. moschiferus* to select 12 major limiting factors for the SDMs [4,10,14], including two bioclimatic variables (mean annual temperature and mean annual precipitation), two topographic variables (elevation and slope), four vegetative variables (birch distribution, larch distribution, shrub distribution, and forest density), and four anthropogenic variables (human modification of terrestrial land, distance from roads, settlement density, and human density) (Table 1). The environmental variables were defined in a raster structure with a cell size of 30”. To avoid bias based on collinearity, we used the variance inflation factor (VIF) to test for collinearity among the selected variables [33]. The VIF was calculated using the BiodiversityR package Version 2.8 in R Version 3.4.3 and was below 10, confirming the independence of the variables (Table 1) [34].

### 2.3. Species Distribution Modeling

We modeled the potential distribution of *M. moschiferus* in Northeast China using ensemble forecasting approaches via the R package BIOMOD2 Version 3.3-7 [35,36]. BIOMOD2 includes ten modeling algorithms and is considered a suitable platform for the ensemble forecasting of species distributions [35,37,38]. Presence–absence models tend to perform better than presence-only models [39]. In this study, we created 2000 random points for pseudo-absence data using geoprocessing tools in ArcGIS Version 10.2.2 [38,40]. To reduce model uncertainty, we first tested all ten modeling algorithms in BIOMOD2 to determine the optimal algorithm, as evaluated by the true skill statistic (TSS) and the area under the curve of the receiver operating characteristic curve [41,42]. Based on these results, three modeling algorithms were selected, including random forest (RF), generalized boosting model (GBM), and multiple adaptive regression splines (MARS). We then tuned the model parameters using the BIOMOD2 package and repeated the analyses 30 times to reduce uncertainty [43,44]. We randomly assigned 80% of the dataset as the training dataset, and model performance was tested with the remaining 20% [35]. Ultimately, 90 modeling evaluation results were obtained (30 replicates of three algorithms), and the average TSS from the 30 replicates of the models was set as the threshold for building an ensemble model [35,37,43]. The habitat suitability maps obtained from models with above-average TSS values were combined to provide an ensemble forecast for *M. moschiferus*. To derive presence–absence distributions from the continuous model outputs of habitat suitability, we applied cutoff values calculated using BIOMOD2, and every output cell was categorized as either present (above the cutoff) or absent (below the cutoff) [43,45,46]. This yielded a potential distribution map for *M. moschiferus*. Since the distribution of species can be limited by land use [47,48], we used land-use data (http://www.resdc.cn/, accessed on 12 November 2020) to remove unsuitable habitat to avoid over-prediction according to habitat selection of the species [4].

### 2.4. Habitat Quality Ranking and Identification of Conservation Priority Areas

We used the core-area zonation algorithm in Zonation 4.0.0 (C-BIG, Helsinki, Finland) to prioritize the landscape for *M. moschiferus* [49,50]. The algorithm involves the iterative removal of the least valuable remaining output cell from BIOMOD2, defined as the smallest aggregate loss of conservation value in accordance with the cell’s contribution to the species distribution [50]. The algorithm identifies core areas to create a priority rank raster ranging from 0 to 1 and scaled by the importance for species conservation (0 represents the lowest priority; 1 represents the highest priority) [50]. We extracted the highest 10% of the cells in terms of habitat quality and defined them as the conservation priority area of *M. moschiferus*. To avoid over-assessment, firstly, we also used land-use data to remove unsuitable habitat according to habitat selection of the species [4]. Then we eliminated all patches less than 28 km^2^ to keep musk deer populations referring to previous research on population density and the movement range of *M. moschiferus* [2,4,5,6,8,51].

### 2.5. Landscape Analysis and Potential Connectivity Corridor Construction

The following three indices were calculated to analyze patch fragmentation using FRAGSTATS Version 4.2 (developed by Kevin Mcgarigal and Eduard Ene) [52]: Patch Cohesion Index (PCI; 0–100), with high PCI indicating a greater concentration of patches and low PCI indicating greater dispersal; Landscape Division Index (LDI: 0–1), which provides a measure of landscape integrity, with low LDI indicating a more connected landscape and high LDI indicating a more fragmented landscape; and Splitting Index (SPLIT: 1—squared number of cells in the landscape), which describes the degree of landscape separation, with 1 representing a single patch and increasing values representing increasing subdivision into smaller patches.

We used the Linkage Mapper GIS tool Version 2.0.0 to identify least-cost paths and pinch points based on high-quality habitat patches and a raster map of resistance [53,54]. A value that reflects the energetic cost, difficulty, and mortality risk when moving across each cell was assigned to the resistance file [53]. In this study, the raster map of resistance was calculated by combining natural resistance (inverted priority rank raster and unsuitable natural land use) and artificial resistance (cultivated land, road, construction land, and orchard land). Least-cost paths provide the best route for the movement of animals between habitat patches [48,53]. Pinch point analysis based on the circuit theory provides corridor resistance values and can be used to identify important areas to keep the entire network connected, which is illustrated by areas with higher current flow in movement pathways [54,55].

## 3. Results

### 3.1. Model Performance, Important Variables and Human Influence on M. moschiferus

On average, the GBM algorithm had the highest TSS value (TSS = 0.794 ± 0.038), followed by MARS (TSS = 0.786 ± 0.037) and RF (TSS = 0.777 ± 0.035). The mean TSS for ensemble models was 0.904 and the mean AUC for ensemble models was 0.988, indicating excellent performance [41]. The most important variables of the *M. moschiferus* distribution were mean annual precipitation and human modification of terrestrial land (Table 1). Human modification of terrestrial land was calculated using 13 anthropogenic stressors reflecting the degree of human influence on biodiversity and ecosystem functioning (range 0–1, where 0 represents the lowest human influence and 1 represents the highest human influence) [56]. Therefore, we assessed the anthropogenic disturbance of *M. moschiferus* habitat using this variable in ArcGIS. The results revealed that core patches in the Greater Khingan Mountains zone had the lowest human influence (0.034), followed by the Lesser Khingan Mountains zone (0.057) and the Changbai Mountains zone (0.127).

### 3.2. Potential Distribution, Conservation Priority Areas, and Conservation Status of M. moschiferus

The model revealed concentrations of *M. moschiferus* in the northern Greater Khingan Mountains, the southeastern Lesser Khingan Mountains, and the eastern Changbai Mountains, and the potential distribution covered an area of 127,442.14 km^2^ (Figure 2). Conservation priority areas included 41 core patches with an area of 106,306.43 km^2^ (Figure 3). The Inner Mongolia Autonomous Region covered 52.87% of total conservation priority areas, followed by the Heilongjiang Province (40.32%), Jilin Province (5.89%), and Liaoning Province (0.92%). Core patches CP1 and CP2 in the northern Greater Khingan Mountains were the two largest core patches in the study area and contained 81.84% of conservation priority areas (Figure 4a). Overall, 15 core patches had an area exceeding 500 km^2^, 13 core patches had an area of 100–500 km^2^, and 13 core patches had an area of 32.28–100 km^2^. These core patches were mainly distributed within ten prefecture-level cities: Hulunbuir in the Inner Mongolia Autonomous Region; Da Hinggan Ling Prefecture, Heihe, Yichun, and Harbin in Heilongjiang Province; Yanbian Korean Autonomous Prefecture, Baishan, and Tonghua in Jilin Province; and Benxi and Dandong in Liaoning Province. In China, national parks and national nature reserves are considered a fundamental backbone for nature conservation [26]. An overlay analysis showed that 39 nature reserves overlapped with the potential distribution of *M. moschiferus* and 30 nature reserves overlapped with priority conservation areas among the 107 national nature reserves in the study area (for detailed information on national nature reserves, see Appendix A). However, these national nature reserves, together with the Northeast Tiger and Leopard National Park, only included 12.46% of the potential distribution and 10.08% of the conservation priority areas of *M. moschiferus* (Figure 2 and Figure 3).

### 3.3. Patch Fragmentation of Core Patches and Potential Connectivity Corridors

Since some adjacent core patches were separated by great distances, we divided the conservation priority areas of *M. moschiferus* into the Greater Khingan Mountains zone, the Lesser Khingan Mountains zone, and the Changbai Mountains zone, according to the biogeographic region in Northeast China. Patch fragmentation mainly occurred in the Changbai Mountains (PCI = 97.774, LDI = 0.887, SPLIT = 8.841) and the Lesser Khingan Mountains (PCI = 98.205, LDI = 0.841, SPLIT = 6.296), with 18 patches and 21 patches, respectively, while conservation priority areas in the Greater Khingan Mountains only contained two core patches, indicative of high connectivity (PCI = 99.963, LDI = 0.107, SPLIT = 1.120) (Figure 3). Linkage Mapper GIS generated 69 potential connectivity corridors connecting 41 core patches (for detailed information on potential connectivity corridors, see Appendix A). Among them, one connectivity corridor with a 2.38 km-long least-cost path was located in the Greater Khingan Mountains zone, 39 connectivity corridors with least-cost paths totaling 1179.948 km in length were located in the Lesser Khingan Mountains zone, and 29 connectivity corridors with least-cost paths totaling 1342.092 km in length were located in the Changbai Mountains zone. Pairwise pinch points analysis indicated that the lowest resistance of movement along the optimal path occurred between the core patches CP8 and CP10 in the Lesser Khingan Mountains zone and that the highest resistance occurred between CP31 and CH35 in the Changbai Mountains zone. In addition, pinch points occurred between almost all adjacent core patches, representing important linkages to keep the network of *M. moschiferus* habitat connected (Figure 4).

## 4. Discussion

Most studies of *M. moschiferus* in China have focused on habitat selection [57,58,59], whereas large-scale studies of the species’ biogeography have only provided a vague distribution range due to low data accuracy [2,60]. In this study, we carried out multiple interviews, questionnaires, and field investigations in the study area from 2019 to 2021 to obtain comprehensive distribution information on *M. moschiferus*. Furthermore, we selected relevant variables according to the species’ ecological requirements, used an ensemble forecasting approach, selected optimal modeling algorithms, tuned the model parameters, and removed unsuitable habitat from the potential distribution map using land-use data to reduce uncertainty. Despite our best efforts, our results may still contain sample biases resulting from reliability of interviews. However, our findings still provide the first assessment of the potential distribution and conservation priority areas of *M. moschiferus* in Northeast China by integrating an ensemble forecasting approach with systematic conservation planning to address conservation needs. On this basis, we built a potential protection network in the three investigated biogeographic regions to support the long-term conservation of the species and to promote gene flow among habitat islands. 

To propose effective conservation decisions, it is essential to determine the cause of a species’ decline [43]. *M. moschiferus* populations and habitats have declined significantly because of poaching and long-term logging [6,61]. After banning hunting and logging in natural forests, habitat fragmentation has become one of the most serious threats to *M. moschiferus* [4,5,62]. However, few studies have assessed potential sites for conservation priority areas and the degree of habitat fragmentation for the species due to limitations in research scale and data accuracy. Musk deer are sensitive to anthropogenic disturbance and prefer primitive forests with little-to-no human activity [6,14,57]. Our results also showed that the human modification of terrestrial land to be an important variable affecting the distribution of *M. moschiferus*.

Our findings revealed that large areas of a high-quality habitat with low habitat fragmentation remain for *M. moschiferus* in the northern Greater Khingan Mountains. In addition, 50.52% of the total occurrence points were collected from the Greater Khingan Mountains, reflecting a large *M. moschiferus* population in this area. Therefore, northern Greater Khingan Mountains may be the most important region to maintain the stability of the *M. moschiferus* population in Northeast China. The Greater Khingan Mountains contain the largest concentrated and contiguous natural forest area in China [28,29]. Therefore, the Greater Khingan Mountains zone could be the first area in which the *M. moschiferus* population will recover in response to hunting and logging bans.

Thanks to the strict conservation efforts and policies of the Chinese government, China’s forests have been recovering over the past three decades [23]. However, forest regions in the Lesser Khingan Mountains and the Changbai Mountains remain highly fragmented and have a high proportion of secondary forest due to excessive, long-term logging [61]. The LDI and SPLIT values demonstrated that *M. moschiferus* core patches in the two zones maintained a high degree of fragmentation. The mean distance of adjacent core patches was 26.93 km in the Lesser Khingan Mountains and 40.03 km in the Changbai Mountains. With the acceleration of infrastructure construction, such as enclosed high-speed rails and expressways, some core patches could become further fragmented. This might prevent the migration of small, isolated populations, leading to localized extinctions. Previous research indicated that the subspecies *M. m. moschiferus* of the Greater Khingan Mountains and *M. m. parvipes* of the Lesser Khingan Mountains were separated 300 km apart from each other in the 1990s, causing geographic isolation [6]. This distance approximately equals the distance from core patch CP2 to CP10 (330 km) in this study. Previous studies did not consider remaining remnant populations in the northwestern Lesser Khingan Mountains (Figure 4b). Nevertheless, the core patches CP3–CP6 of this region have become habitat islands due to their large distance to the meta-population in the southeastern Lesser Khingan Mountains. Therefore, linkages between CP6 and CP10 are important to ensure a gene flow of *M. m. parvipes* in the Lesser Khingan Mountains.

In the Changbai Mountains, the population of *M. m. parvipes* dropped from 3000 individuals in the 1970s to 150 individuals in the 2000s, and the distribution range was reduced to isolated patches with lower human influence [6]. Resistance values for constructing potential connectivity corridors in this region far exceeded those of the other regions; the resistance value of the Changbai Mountains was twice as high as that of the Lesser Khingan Mountains and 291 times higher than that of the Greater Khingan Mountains (Appendix A). Pinch point analysis revealed widespread bottlenecks in the movement pattern of *M. m. parvipes* (Figure 4c). These are critical linkage nodes for the maintenance of a connectivity network. Such pinch points could be the result of natural and artificial resistance, and should be evaluated in an additional field survey. Nevertheless, we conclude that it is crucial to connect the Northeast China Tiger and Leopard National Park and the Changbai Mountain National Nature Reserve. Despite the two regions having the highest biodiversity in Northeast China, the gene flow of terrestrial species between the two regions has been limited due to long-term human development driving land-use changes [63]. Along the path of lowest resistance (linkages 26–33, 33–34), Amur tigers and leopards in the national park could also migrate to recovered areas in the Changbai Mountain National Nature Reserve that have become suitable for habitation, which might further help to meet the goal of dispersing tiger and leopard populations. Considering this, we referred to previous research on Amur tigers to set 20 km in width to serve the needs of more kinds of animals [64].

There are 107 national nature reserves and one national park established in our study area. However, these protected areas only include 10.08% of the conservation priority areas of *M. moschiferus* and offer limited protection to the species, as they are far from covering all core patches. This situation may improve in the future, given the Chinese government’s recent support for the integration and optimization of natural protected areas to solve the incongruous spatial distribution of protected areas versus the distribution of biodiversity; in particular, the government has focused on promoting the establishment of protected areas with national parks [26]. National parks comprise areas that showcase ecosystems characteristic of China, with the purpose of reducing fragmentation and to achieving complete protection of large-scale ecosystems [21]. In 2020, the Chinese government proposed entry criteria for national parks and put forward a flagship species and eco-geographical regions list (China National Standard; ID: GB/T 39737-2020). Compared to flagship species with narrow distributions in Northeast China, such as *Panthera tigris* or *Panthera pardus*, the habitat of *M. moschiferus* spans almost the entire forest ecosystem in Northeast China, including coniferous forests of the north Greater Khingan Mountains, coniferous and broad-leaved mixed forests of the Greater and Lesser Khingan Mountains, and coniferous and broad-leaved mixed forests of the Changbai Mountains. Considering the high habitat quality requirements and endangered status of *M. moschiferus*, we propose that the Chinese government accelerates the construction of the Greater Khingan Mountains National Park and the Lesser Khingan Mountains National Park, and enlarges the Northeast China Tiger and Leopard National Park to improve conservation of endangered species including *M. moschiferus* and address the fragmentation of protected areas and species habitat. In addition, we propose that the Chinese should strengthen international cooperation with neighboring countries including Russia and North Korea.

## 5. Conclusions

We identified that an ecological network for *M. moschiferus* in Northeast China composed of a total of 41 core patches and 69 linkages by combining SDMs, a systematic conservation planning algorithm, and the circuit theory. We conclude that large areas of high-quality *M. moschiferus* habitat with low habitat fragmentation and low human influence remain in the northern Greater Khingan Mountains. In contrast, the habitat in the Lesser Khingan Mountains and the Changbai Mountains showed a high degree of fragmentation and high human influence. Although the Chinese government has listed *M. moschiferus* as a national first-class protected animal and a flagship species, most of its potential distribution and conservation priority areas have not been incorporated into existing protected area systems. Therefore, large-scale protected areas, such as national parks, must be constructed in important eco-geographical regions to support the conservation of this endangered species in China. In the future, we will further explore the detailed cause of pinch points in potential connectivity corridors via field surveys to support construction of ecological corridors within the study area.

## Figures and Tables

**Figure 1 animals-12-00260-f001:**
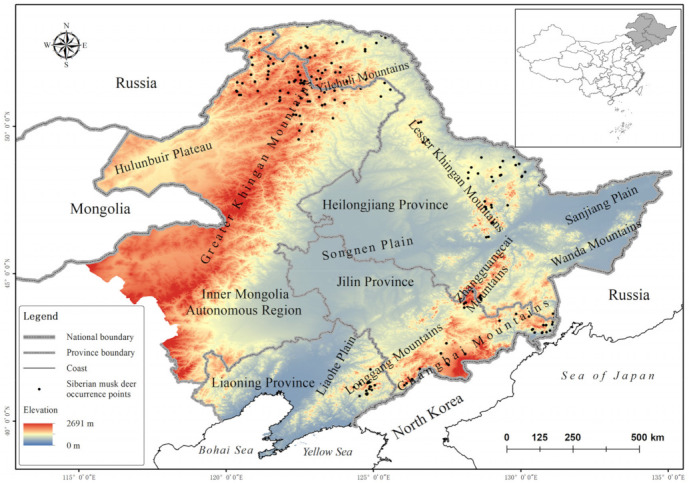
Occurrence points of Siberian musk deer (*Moschus moschiferus*) in Northeast China.

**Figure 2 animals-12-00260-f002:**
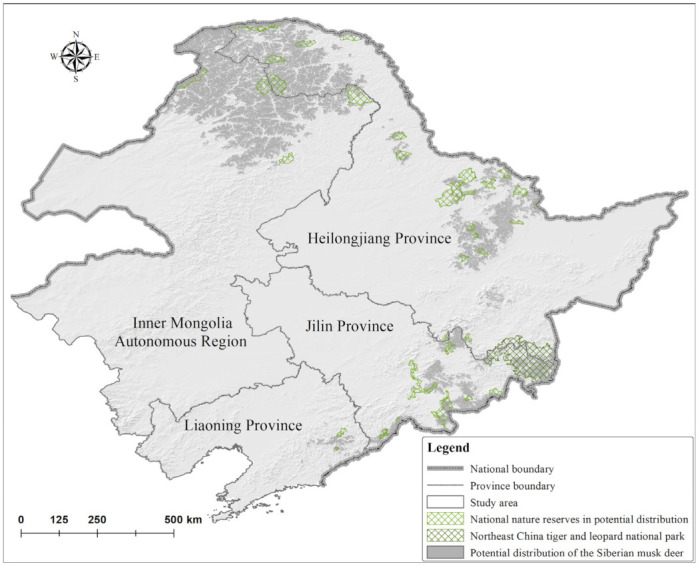
Potential distribution and conservation status of *M. moschiferus* (For basic information on protected areas, see Appendix A: Appendix A).

**Figure 3 animals-12-00260-f003:**
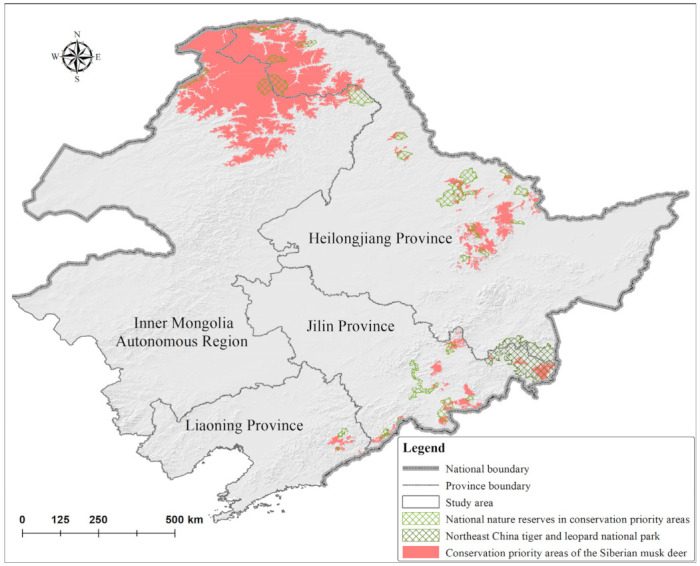
Conservation priority areas and conservation status of *M. moschiferus* (For basic information on protected areas, see Appendix A: Appendix A).

**Figure 4 animals-12-00260-f004:**
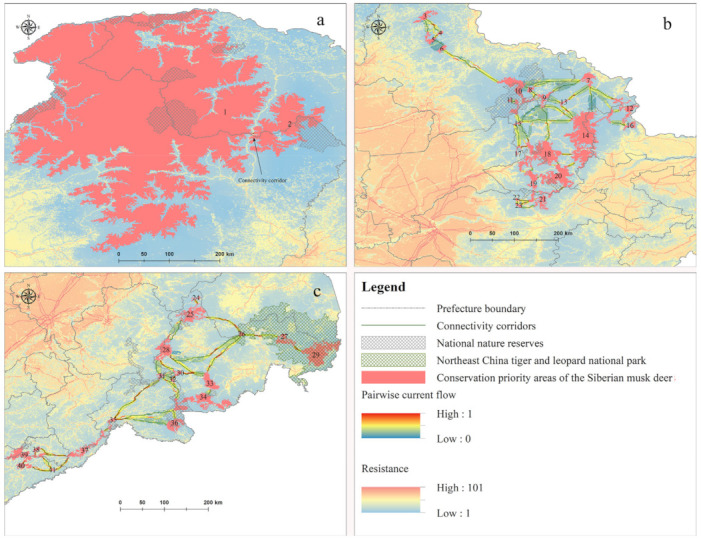
Potential connectivity corridor of *M. moschiferus* in three biogeographic regions ((**a**): Greater Khingan Mountains zone; (**b**): Lesser Khingan Mountains zone; (**c**): Changbai Mountains zone; corridor width: 20 km).

**Table 1 animals-12-00260-t001:** Variance inflation factor value, environmental variable importance, and data sources.

Variables	VIF	Variable Importance	Source
Mean annual temperature	6.261	0.052	(http://www.worldclim.org/, accessed on 23 May 2020)
Mean annual precipitation	2.644	0.237	(http://www.worldclim.org/, accessed on 23 May 2020)
Elevation	3.328	0.015	(http://www.cgiar-csi.org/, accessed on 16 November 2020)
Slope	1.044	0.021	(http://www.cgiar-csi.org/, accessed on 16 November 2020)
Birch distribution	2.854	0.012	(http://www.resdc.cn/, accessed on 8 December 2017)
Larch distribution	2.118	0.022	(http://www.resdc.cn/, accessed on 8 December 2017)
Shrub distribution	3.177	0.005	(http://www.resdc.cn/, accessed on 8 December 2017)
Forest density	4.095	0.046	(http://www.resdc.cn/, accessed on 8 December 2017)
Human modification of terrestrial land	3.145	0.237	(https://sedac.ciesin.columbia.edu/, accessed on 1 October 2019)
Distance from roads	1.508	0.002	(https://www.webmap.cn/, accessed on 7 September 2021)
Settlement density	3.442	0.053	(https://www.webmap.cn/, accessed on 7 September 2021)
Human density	1.295	0.006	(http://www.resdc.cn/, accessed on 8 October 2020)

Note: VIF: Variance inflation factor; variable importance was calculated by species distribution modeling.

## Data Availability

Not applicable.

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
