# Peer review of "Identification of Conservation Priority Areas and a Protection Network for the Siberian Musk Deer (Moschus moschiferus L.) in Northeast China"

_animals, 2022, doi:10.3390/ani12030260_

Round 1

Reviewer 1 Report

To identify an efficiently operating protection network following has to/can be also regarded in the research and result assessment:

In datasests:

1) the knowledge about the approximate number of individual animals participating in migrations

In results assessment:

2) what is the maximum lenght of coridors allowed between two neighbouring core patches (very long corridors must be eliminated from the network - they are not safe for animals; if a corritor is too long - an artificial core patch must be constructed/planted on the route to reduce the excessive lenght of corridor)?

3) what is the minimum size of the core patch able to keep local animal population (smaller than recommended minimum must be removed from the network)

In landscape planning:

4) identification of suitable sites for ecoduct construction over speed railways and expressways could be a very practical contribution/output for the transportation network construction/completing

References:

MacARTHUR, R. H., WILSON, E. O. (1967): The Theory of Island Biogeography. Princeton University Press, Princeton

Reviewer 2 Report

I find this manuscript very well written and the work very well designed and analyzed. My only suggestion is the refer to the conservation situation of the species across the national border. Since your study area borders three different countries, and much of the possible distribution area and many of the important conservation areas are near the international borders, it might be helpful for the readers to know whether there are also protected areas across the borders (possibly for the same populations and/or subspecies) and whether your suggested conservation priority areas and current and suggested NPs can be also connected to areas across the national borders and thus provide even larger PA for the recovery of the species. 

Reviewer 3 Report

The authors have produced an interesting and useful manuscript, containing information necessary for designing protected areas for the conservation of the threatened Siberian musk deer. Their manuscript is very well written, analyzed and discussed. Some clarifications are needed to become more understandable to the reader and ready for publication.

Comments

---Lines 62-63 – coniferous and broad-leaved mixed forests…

---Lines 117-119 – Were the authors able to check the reliability of sightings reported by local residents? How did interviews differ from questionnaires? What kind of literature sources were used? Scientific, local press, other? Such information should be included here for the reader to be able to assess data reliability. Limitations in data collection should be discussed in the Discussion section.

---Line 137 – Table caption should precede Table. Explain in a table footnote what “VIF” stands for. Also, explain that environmental variable importance was assessed by species distribution modeling. The reader should understand information in a table without having to resort to the text.

---Tables 2, 3, 4 – Although tables are fine, the differences between Table 2 and Table 3 are not easily discernible, and also corridors in Table 4 could be better presented. I propose to: a) Add in the caption of Figure 3 “see Table 4 for more detailed maps, b) provide maps of Figure 4 in page-wide length each for better visibility and add information about corridor width in the caption. See also next comment.

---Lines 240-244 – No information about corridor width has been provided. This is critical for corridor use by a species. What are the species’ requirements? How have the authors provided for this? Please give relevant information and discuss it in the Discussion section, along with possible limitations or future research needs.

---Lines 256-258 – Comment here about the accuracy of methodology, as in the Methods section.

---Lines 277-283, 285-289 – Move to results. Use text without numbers here.

---Lines 345, 346 – coniferous and broad-leaved mixed forests…

Round 2

Reviewer 1 Report

Do not forget the maximum lenght of corridors in your further research please. The widht of corridors is also important, but the maximum lenght much more.